# The Flexibility Trap: Rethinking the Value of Arbitrary Order in Diffusion Language Models

Zanlin Ni[1]   Shenzhi Wang[1]   Yang Yue[1]   Tianyu Yu[2]   Weilin Zhao[2]   Yeguo Hua[3]
Tianyi Chen[3]   Jun Song[4]   Cheng Yu[4]   Bo Zheng[4]   Gao Huang[1][5]

## Abstract

Diffusion Large Language Models (dLLMs) break the rigid left-to-right constraint of traditional LLMs, enabling token generation in arbitrary orders. Intuitively, this flexibility implies a solution space that strictly supersets the fixed autoregressive trajectory, theoretically unlocking superior reasoning potential. However, in this paper, we find that for general reasoning tasks (*e.g.*, mathematics and coding), arbitrary order generation may in fact limit the reasoning potential of dLLMs. We observe that dLLMs tend to exploit this order flexibility to bypass high-uncertainty tokens that are crucial for exploration, which can lead to a premature collapse of solution coverage. This observation motivates a rethink of RL approaches for dLLMs, where considerable complexities, such as handling combinatorial trajectories and intractable likelihoods, are often devoted to preserving this flexibility. We show that effective reasoning can be elicited by simply forgoing arbitrary order and applying standard Group Relative Policy Optimization (GRPO) instead. Our approach, **JustGRPO**, is minimalist yet surprisingly effective (*e.g.*, 89.1% accuracy on GSM8K) while fully retaining the parallel decoding ability of dLLMs. Code: https://github.com/LeapLabTHU/JustGRPO.

## 1. Introduction

Recent research has witnessed a surge in Diffusion Large Language Models (dLLMs) (Nie et al., 2025; Ye et al.,

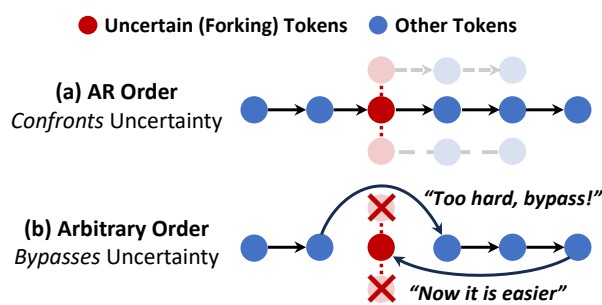

*Figure 1.* **Confronting *vs.* bypassing uncertainty.** (a) **AR order** preserves reasoning space by forcing decisions at uncertain tokens. (b) **Arbitrary order** bypasses uncertainty and resolves easier tokens first. Once future context is established, the original branching possibilities are effectively pruned.

2025b; Zhu et al., 2025; Zhao et al., 2025; Inception Labs et al., 2025), which challenge the dominant autoregressive (AR) paradigm (Brown et al., 2020; Achiam et al., 2023) by treating sequence generation as a discrete denoising process. Central to the appeal of dLLMs is their theoretical flexibility, which offers two distinct advantages over the strict left-to-right causal chain of AR models: efficient parallel decoding and the capability for arbitrary-order generation.

While the efficiency gains of parallel decoding have been well-established (Wu et al., 2026b; Inception Labs et al., 2025; DeepMind, 2025; Song et al., 2025; Wu et al., 2026a), the implications of arbitrary-order generation remain relatively less explored. Theoretically, the unconstrained generation order constitutes a superset of the fixed autoregressive trajectory and has the potential to unlock more advanced problem-solving abilities. Meanwhile, some works (Ye et al., 2025a; Kim et al., 2025) have demonstrated the superiority of non-sequential generation for specific tasks such as sudoku and zebra puzzles. Driven by these successes and the theoretical promise, the community has increasingly adopted Reinforcement Learning (RL) to elicit similar advanced capabilities for *general reasoning tasks* such as mathematics and coding[1] (Zhao et al., 2025; Wang et al.,

[1]LeapLab, Tsinghua University [2]NLPLab, Tsinghua University [3]Tsinghua University [4]Alibaba Group [5]BNRist, Tsinghua University. Correspondence to: Gao Huang <gaohuang@tsinghua.edu.cn>.

*Proceedings of the 43rd International Conference on Machine Learning*, Seoul, South Korea. PMLR 306, 2026. Copyright 2026 by the author(s).

---

[1]We focus on these domains as the primary representatives of *general reasoning* due to their broad practical utility and central role in benchmarking the core intelligence of reasoning models.

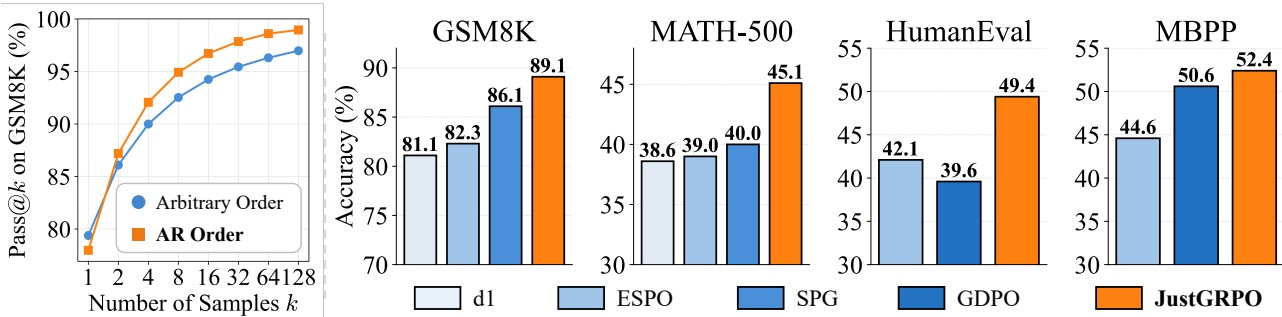

*Figure 2.* **Less flexibility unlocks better reasoning potential.** *Left*: We observe a counter-intuitive phenomenon where restricting dLLMs to standard Autoregressive (AR) order expands the reasoning solution space. *Right*: Motivated by this, we propose "**JustGRPO**". By foregoing complex arbitrary-order adaptations and adopting just GRPO, we elicit the reasoning capability of dLLMs more effectively. Results reported on LLaDA-Instruct (Nie et al., 2025) with generation length of 256. Baseline results are quoted from their original papers. We also provide our reproduced baselines in Table 2.

2026a; Gong et al., 2026; Rojas et al., 2026; Ou et al., 2026).

In this work, we present a counter-intuitive observation: for these general reasoning tasks, arbitrary-order generation may in fact narrow rather than expand the reasoning potential elicitable by RL. To rigorously assess this, we employ Pass@$k$ (Chen et al., 2021), which measures the coverage of solution space. Recent studies suggest that RL primarily acts to sharpen the base model's distribution; consequently, the Pass@$k$ performance of the base model serves as an effective proxy for the achievable reasoning limits of the model after RL training (Yue et al., 2025; Liu et al., 2025). Under this metric, we compare the reasoning potential of arbitrary-order generation against standard AR decoding using LLaDA-Instruct (Nie et al., 2025). As shown in Figure 2 (Left), restricting a dLLM to standard AR order tends to yield a *higher* Pass@$k$, and consequently a higher reasoning boundary, than its flexible counterpart.

This counter-intuitive observation is closely related to how different orders handle uncertainty. Reasoning is inherently non-uniform: it hinges on sparse "forking tokens", typically connectives like "Therefore" or "Since", which do not merely continue a sentence but steer the logical trajectory into distinct branches (Wang et al., 2025; Cheng et al., 2026; Huang et al., 2025a). At these forks, the reasoning path diverges, naturally manifesting as localized spikes in entropy (Wang et al., 2025). Standard AR decoding compels the model to *confront* this uncertainty (Figure 1a). By sampling exactly at these forks, the model is able to explore different reasoning paths, thereby preserving the diversity of the generated rationales. Arbitrary-order generation, however, allows the model to *bypass* these hard decisions (Figure 1b). This flexibility is typically exploited to prioritize low-uncertainty tokens (Nie et al., 2025), which effectively anchors the reasoning path on easier parts before deciding the connectives that lead to them. By the time the model returns to infill the bypassed forks, the established future

context has prematurely resolved their ambiguity. We term this phenomenon *entropy degradation*: the high entropy originally inherent to the fork is suppressed. Effectively, the model is no longer navigating a branching point, but aligning the connective to bridge a pre-determined gap.

The above observations motivate a rethink of RL for dLLMs in these general reasoning tasks. Current methods operate under the assumption that preserving arbitrary-order flexibility is essential. This commitment incurs a heavy tax: algorithms must grapple with a combinatorial explosion of denoising trajectories (Zhao et al., 2025; Yang et al., 2025; Gong et al., 2026) and intractable marginal likelihoods (Ou et al., 2026; Wang et al., 2026a), forcing reliance on inaccurate approximations (Wang et al., 2026a; Ou et al., 2026; Rojas et al., 2026). However, if arbitrary order is non-essential, or even detrimental for eliciting the potential for the target tasks, this complexity may be unnecessary.

To this end, we propose a return to simplicity with **Just-GRPO**. We show that eliciting reasoning potential for general reasoning tasks does not necessarily require complex, diffusion-specific RL adaptations. Instead, it can be effectively achieved by simply treating the dLLM as an AR policy during RL training. This allows us to apply standard GRPO (Shao et al., 2024) without modification, transforming an otherwise intractable optimization into a well-defined task. Crucially, this AR formulation only serves as a "scaffold" during RL training for better exploration. The dLLM's native architecture remains untouched, *e.g.*, no causal masking imposed, which preserves their bidirectional attention and discrete diffusion formulation. As a result, the *elicitation of reasoning ability* (which benefits from sequential exploration) and the *inference execution* (which benefits from parallel decoding of dLLMs) are decoupled.

Despite its simplicity, JustGRPO is surprisingly effective. It achieves strong results (*e.g.*, 89.1% accuracy on GSM8K, 45.1% on MATH-500), comparing favorably with meth-

ods relying on intricate diffusion-specific RL adaptations (Table 1). More importantly, as shown in Figure 8, the AR-trained model remains fully compatible with parallel decoding, suggesting that the reasoning capabilities acquired via AR exploration need not compromise the signature inference efficiency of dLLMs.

## 2. Related Work

**Diffusion language models.** Motivated by the success of diffusion models in continuous domains (Ho et al., 2020; Rombach et al., 2022), recent work has extended diffusion to discrete text generation. Early embedding-space approaches (Li et al., 2022; Gong et al., 2023; Han et al., 2023) faced optimization and discretization challenges, leading to the adoption of masked diffusion models (Lou et al., 2024; Sahoo et al., 2024; Shi et al., 2024; Ou et al., 2025), which operate directly in the discrete token space via random masking. Recent large-scale models like LLaDA (Nie et al., 2025) and Dream (Ye et al., 2025b) achieve performance competitive with autoregressive (AR) models. Among various advantages of dLLMs discussed in the literature, two aspects have received particular attention. First, dLLMs naturally support parallel decoding, enabling significant inference acceleration (Wu et al., 2026b; Ben-Hamu et al., 2025; Inception Labs et al., 2025; DeepMind, 2025; Song et al., 2025). Second, their non-autoregressive formulation allows for arbitrary-order token generation, which has been hypothesized to benefit complex reasoning by relaxing strict left-to-right constraints (Ye et al., 2025a; Kim et al., 2025).

**The value of order arbitrariness.** While early studies validated arbitrary order generation in constrained tasks (Ye et al., 2025a; Nie et al., 2025; Kim et al., 2025), recent research extends this to general reasoning. Some works demonstrate that dLLMs can naturally exhibit non-standard decoding behaviors (*e.g.*, sketch-first) (Xie et al., 2025; Gong et al., 2026), while others explicitly optimize decoding schedules to boost performance (Peng et al., 2025; Huang et al., 2025b). Moreover, some studies (Gong et al., 2026; Hong et al., 2026; Shen et al., 2026) also employ Pass@$k$ to measure the reasoning potential of dLLMs. Recognizing this potential, these works predominantly focus on enhancing performance within the arbitrary-order paradigm, leaving less-examined whether the benefit is uniquely tied to the order arbitrariness. Recently, (Du et al., 2025) challenged the necessity of arbitrary order, arguing that forcing models to learn uniform permutations causes a significantly looser approximation of the underlying data distribution. While their analysis focuses on pre-training, we observe a parallel failure mode in reasoning and RL, suggesting that the flexibility can degrade exploration essential for RL training. This is also related to the broader view (Varshney & Goyal, 2006) in which the ordering of a sequence carries useful structural information beyond its content, which arbitrary-order generation tends to overlook.

**Reinforcement learning for diffusion language models.** Reinforcement learning for dLLMs faces structural optimization hurdles distinct from the autoregressive paradigm, primarily stemming from the combinatorial explosion of denoising trajectories. While early attempts (Zhao et al., 2025; Yang et al., 2025; Gong et al., 2026) sought to adapt token-level formulations directly, they were often limited by ill-defined state transitions, necessitating reliance on mean-field approximations. Consequently, the field has shifted toward sequence-level perspectives (Zhu et al., 2025; Wang et al., 2026a; Rojas et al., 2026; Ou et al., 2026), employing various surrogates to approximate the intractable marginal likelihood. Yet, an off-policy misalignment often persists across these methods: the heuristic-guided sampling required for efficient exploration diverges from the underlying diffusion prior, which can bias the gradients without principled correction (Schulman et al., 2015). Two notable exceptions partially address these issues. LLaDOU (Huang et al., 2025b) introduces an auxiliary policy to model token position selection, enabling direct trajectory likelihood estimation, while TraceRL (Wang et al., 2026b) aligns optimization with inference traces via a shrinkage-based stepwise MDP. Nevertheless, both retain the full arbitrary-order mechanism, implicitly assuming its necessity, whereas we question whether effective RL training can be achieved more simply by reconsidering the need for arbitrary order itself.

## 3. Preliminaries

### 3.1. Diffusion Large Language Models

Diffusion Large Language Models (dLLMs), particularly Masked Diffusion Models (MDMs), generate sequences by iteratively denoising a masked state $x_t$ initialized from fully masked tokens. The process is indexed by a continuous time variable $t \in [0, 1]$, representing the masking ratio. Given a clean sequence $x_0$, the forward process independently masks each token with probability $t$:

$$q(x_{t,k} \mid x_{0,k}) = \begin{cases} \texttt{[MASK]}, & \text{with prob } t, \\ x_{0,k}, & \text{with prob } 1 - t, \end{cases} \quad (1)$$

where $k$ is the token index. A neural network $p_\theta(x_0 \mid x_t)$ estimates the original token distribution at masked positions. During inference, generation starts from $x_1$ (all $\texttt{[MASK]}$) and iteratively unmasks a subset of tokens based on heuristics such as confidence scores, updating $x_t \rightarrow x_{t-\Delta t}$ until completion. As a special case, dLLMs can also perform generation autoregressively by always unmasking the leftmost token. The model is trained by minimizing the Negative Evidence Lower Bound, which reduces to a weighted cross

entropy loss over masked tokens:

$$\mathcal{L}_{\text{MDM}}(\theta) = -\mathbb{E}_{t \sim \mathcal{U}[0,1], \, x_t \sim q(x_t | x_0)}$$

$$\left[ \frac{1}{t} \sum_{k=1}^{L} \mathbf{1}[x_{t,k} = [\text{MASK}]] \log p_\theta(x_{0,k} \mid x_t) \right]. \quad (2)$$

### 3.2. Group Relative Policy Optimization

Group Relative Policy Optimization (GRPO) is a reinforcement learning algorithm that avoids value function estimation by using group level reward normalization. It is typically applied to autoregressive policies $\pi_\theta(o_k \mid o_{<k}, q)$. For each query $q$, GRPO samples a group of $G$ outputs $\{o_1, \ldots, o_G\}$ from the old policy $\pi_{\theta_{\text{old}}}$. An advantage $\hat{A}_{i,k}$ is computed by standardizing the reward $r(o_i)$ against the group statistics: $\hat{A}_{i,k} = (r(o_i) - \mu_G)/\sigma_G$, where $\mu_G$ and $\sigma_G$ are the group mean and standard deviation. The GRPO objective maximizes a clipped surrogate function with a KL regularization term:

$$\mathcal{J}(\theta) = \mathbb{E}_{q \sim P(Q), \{o_i\}_{i=1}^G \sim \pi_{\theta_{\text{old}}}} \left[ \frac{1}{G} \sum_{i=1}^{G} \frac{1}{|o_i|} \sum_{k=1}^{|o_i|} \right.$$

$$\left. \left( \min \left( \rho_{i,k} \hat{A}_{i,k}, \text{clip}(\rho_{i,k}, 1 - \epsilon, 1 + \epsilon) \hat{A}_{i,k} \right) - \beta \mathbb{D}_{\text{KL}} \right) \right]$$

$$(3)$$

with the token level importance ratio $\rho_{i,k} = \frac{\pi_\theta(o_{i,k} | o_{i,<k}, q)}{\pi_{\theta_{\text{old}}}(o_{i,k} | o_{i,<k}, q)}$.

### 3.3. Pass@$k$ as a Proxy for Reasoning Potential

To rigorously quantify the reasoning capability boundaries of dLLMs, we adopt the Pass@$k$ metric (Chen et al., 2021; Yue et al., 2025). In the context of Reinforcement Learning with Verifiable Rewards (RLVR), which is currently the dominant paradigm for enhancing reasoning capabilities, exploration is a prerequisite for improvement. An RL agent can only reinforce correct reasoning paths if it is capable of sampling them during the exploration phase to obtain a positive reward signal.

Accordingly, Pass@$k$ has been widely established as a standard measure of a model's *reasoning potential* (Yue et al., 2025; Liu et al., 2025; Zhang et al., 2025a). It measures the probability that at least one correct solution is generated within $k$ independent samples, effectively representing the upper bound of the solution space accessible to RL optimization. Formally, following the unbiased estimator formulation (Chen et al., 2021; Yue et al., 2025), given $n$ samples where $c$ are correct, Pass@$k$ is calculated as:

$$\text{Pass@}k = \mathbb{E} \left[ 1 - \frac{\binom{n-c}{k}}{\binom{n}{k}} \right] \quad (4)$$

A high Pass@$k$ indicates that the correct reasoning trajectory lies within the model's sampling distribution and is thus learnable via RL optimization. Conversely, if a model consistently fails to yield a solution despite a vast sampling budget, it suggests the problem lies beyond its intrinsic reasoning boundary. In such scenarios, standard RLVR methodologies are fundamentally limited by the absence of positive exploration signals (Yue et al., 2025).

## 4. The Flexibility Trap

In this section, we empirically test whether the flexibility of arbitrary-order generation translates into higher reasoning potential. We compare two decoding modes: *Arbitrary Order*, which follows standard diffusion decoding with low-confidence remasking (Nie et al., 2025; Zhao et al., 2025; Wu et al., 2026b), and *AR Order*, where arbitrary-order flexibility is disabled and generation is strictly constrained to left-to-right decoding. We adopt the commonly used experimental setup in prior diffusion LLM work: a maximum of 256 tokens decoded over 256 steps, a semi-autoregressive block size of 32. Sampling temperature is set to 0.6 as in (Yue et al., 2025). The prompt template follows (Zhao et al., 2025). Results under alternative temperatures and sampling configurations are deferred to Appendix B.

### 4.1. Arbitrary Order Can Limit Reasoning Potential

**Pass@$k$ analysis.** We first evaluate the reasoning potential using the Pass@$k$ metric on three representative dLLMs: LLaDA-Instruct (Nie et al., 2025), Dream-Instruct (Ye et al., 2025b), and LLaDA 1.5 (Zhu et al., 2025) on four reasoning benchmarks: GSM8K, MATH-500, HumanEval, and MBPP. As shown in Figure 3, while arbitrary order often achieves competitive and in many cases better performance at $k = 1$, it exhibits a notably flatter scaling curve compared to the AR mode. As $k$ increases, the AR mode demonstrates a stronger capability to uncover more correct solutions.

**Solution space coverage.** One might wonder whether arbitrary order explores a different solution space, albeit less efficiently, which could account for its lower reasoning potential. We test this by analyzing solution coverage at $k = 1024$ using LLaDA-Instruct. As shown in Figure 4, we find that solutions obtained via arbitrary-order generation largely overlap with those discovered by AR decoding, and in practice form a smaller subset. For instance, on HumanEval, 21.3% of problems are solved exclusively by AR, while the converse case remains rare (0.6%). Although arbitrary-order generation theoretically implies a larger solution space, the set of solutions effectively reachable under practical sampling regimes appears more constrained.

**More arbitrariness, less potential.** So far we have contrasted arbitrary order with AR order as two extremes,

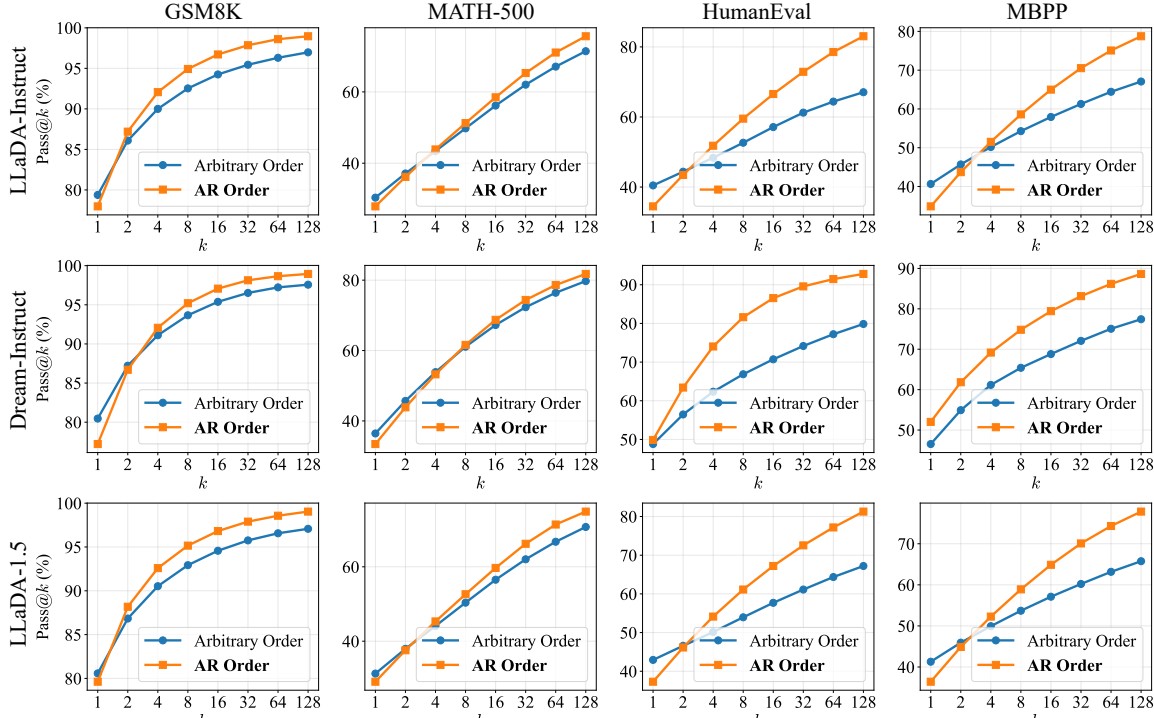

Figure 3. **Reasoning potential measured by Pass@$k$.** While arbitrary order is competitive in single-shot settings ($k = 1$), it exhibits flatter scaling curves compared to AR Order. This observation is consistent across different dLLMs and general reasoning benchmarks.

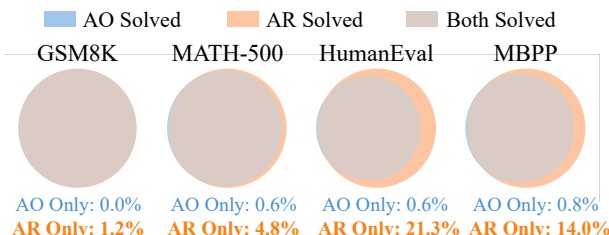

Figure 4. **Solution space coverage** measured by Pass@1024. The problems solvable by arbitrary order (AO) are largely a subset of those solved by AR Order. Results measured with LLaDA-Instruct.

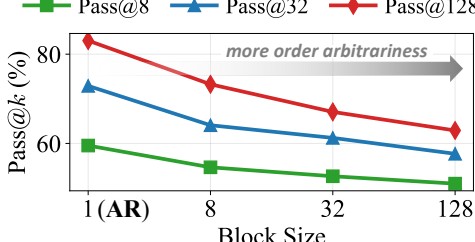

Figure 5. **More arbitrariness, less potential.** We sweep the semi-autoregressive decoding block size $B$ for different levels of order arbitrariness: $B = 1$ recovers pure AR order and larger $B$ grants the sampler more freedom in choosing which positions to resolve next. Results are measured on HumanEval with LLaDA-Instruct.

but the degree of arbitrariness is not binary. The semi-autoregressive protocol in dLLMs (Nie et al., 2025; Ye et al., 2025b) controls it through the block size $B$: the sequence is split into blocks of size $B$, and within each block the model adaptively chooses tokens to unmask. Thus $B$ sets the range within which the model can freely choose the next position, with $B = 1$ leaving no freedom (pure AR order) and larger $B$ allowing more arbitrary decoding; our experiments above use $B = 32$, following (Zhao et al., 2025). We now sweep $B$ to examine how the degree of arbitrariness affects reasoning potential.

As shown in Figure 5, the trend is consistent and monotonic: across every $k \in \{8, 32, 128\}$, Pass@$k$ decreases as $B$ grows. This suggests the effect is consistent rather than tied to the two extreme settings: in this setting, less arbitrariness

is consistently associated with higher reasoning potential.

### 4.2. Mechanism: The Entropy Degradation

**Adaptive decoding bypasses logical forks.** To understand why the theoretically larger solution space of dLLMs appears to collapse in practice, we examine more closely how the two decoding modes behave. In AR order, the model is constrained to strictly resolve the left-most unknown token at each step, forcing the model to confront uncertainty as it arises. In contrast, arbitrary order *adaptively* selects tokens to update based on model confidence, preferentially generating "easy" tokens with high certainty while bypass-

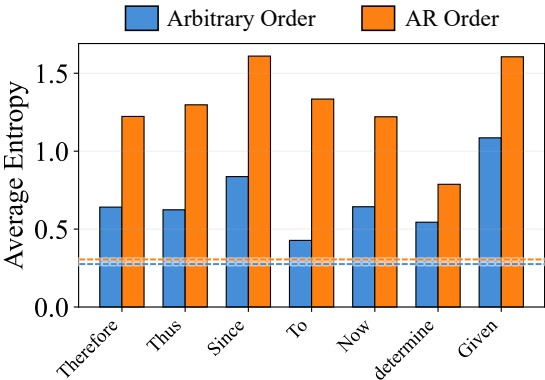

*Figure 6.* **Frequently bypassed tokens** in arbitrary order are typically logical connectors and transition words. Results are measured on MATH-500 with LLaDA-Instruct.

*Figure 7.* **Entropy degradation.** While the global average token entropy (dashed lines) of arbitrary order remains comparable to AR, the entropy at logical forks (blue bars) drops significantly. Results are measured on MATH-500 with LLaDA-Instruct.

ing "hard" ones. Inspecting the frequently bypassed tokens reveals a clear pattern: As shown in Figure 6, the diffusion sampler disproportionately defers logical connectives and transition markers such as "Therefore", "Thus", and "Since". Prior work has shown that such tokens often have high entropy (which also holds true in dLLMs; see Figure 7), and act as "reasoning sparks" or "logical forks", functioning as branching points that determine subsequent reasoning directions (Wang et al., 2025; Cheng et al., 2026; Huang et al., 2025a). In conventional language models, keeping these tokens in high-entropy state is critical for effective exploration of the reasoning space (Wang et al., 2025).

**The "entropy degradation" phenomenon.** How does the adaptive behavior of arbitrary order affect these logical forking tokens? We measure the entropy of them when they are decoded in Figure 7 (more results in Appendix B.3). In AR order, these tokens generally maintain high entropy when decoded, indicating a relatively fruitful branching point where multiple reasoning paths remain viable (Wang et al., 2025). In contrast, arbitrary order exhibits a sharp decrease in entropy. By deferring the logical connectors, the model prioritizes generating easier future tokens *before* deciding the logic connections that lead to them. When the model

eventually returns to fill in the bypassed connectors, the presence of future context significantly reduces uncertainty. As a result, the model is no longer making an open-ended navigational decision at a fork, but more like a retrospective alignment to bridge the gap to a pre-determined conclusion. We term this phenomenon *entropy degradation*.

**Conclusion.** In summary, our findings suggest that the flexibility of arbitrary order in general reasoning tasks may inadvertently favor inference-time exploitation over exploration. The observed entropy degradation serves as a quantitative indicator of this tendency: by prioritizing low-uncertainty decisions, the model implicitly anchors its trajectory to specific outcomes early in the generation process. This effectively constrains the solution space, potentially yielding higher single-shot coherence but limiting the broad exploration required for complex problem-solving. Standard AR ordering, by strictly adhering to the causal chain, retains the high-entropy nature of logical forks. This inability to circumvent critical decision points encourages the model to sample from diverse reasoning branches, which helps preserve the solution coverage essential for reinforcement learning.

## 5. "Just GRPO" for dLLMs

The findings in Section 4 suggest that arbitrary order can limit the reasoning potential accessible to RL. Despite this, current RL methods for dLLMs remain heavily burdened by the need to preserve this specific flexibility. In this section, we examine the "tax" imposed by this flexibility (Section 5.1) and show that forgoing it enables a minimalist yet effective solution: **JustGRPO** (Section 5.2).

### 5.1. The Flexibility Tax in dLLMs' RL

Existing diffusion RL methods operate under the premise that the policy must accommodate the combinatorial space of denoising trajectories $\mathcal{T}$ to preserve arbitrary order generation. This design choice, while conceptually general, introduces several challenges:

**Ambiguity in token-level decomposition.** In dLLMs, a generation state $s_t$ is a noisy sequence conditioned on a stochastic unmasking trajectory $\tau$. Unlike autoregressive models, dLLMs do not admit a unique, index-aligned conditional probability of the form $\pi(o_t \mid s_t)$, making token-level credit assignment ambiguous and rendering the standard importance ratio $\rho_t = \frac{\pi_\theta(o_t|s_t)}{\pi_{\text{old}}(o_t|s_t)}$ hard to define.

**Intractable sequence likelihood.** Autoregressive models factorize the sequence likelihood as $\log \pi(o) = \sum_t \log \pi(o_t \mid o_{<t}, q)$, whereas dLLMs require marginalization over all valid denoising trajectories, $\pi_\theta(o \mid q) = \sum_{\tau \in \mathcal{T}} \pi_\theta(o, \tau \mid q)$. For a sequence of length $N$, the trajectory space grows as $|\mathcal{T}| = O(N!)$, rendering exact likeli-

hood computation infeasible and forcing existing methods to rely on ELBO-based surrogates rather than the original objective (Wang et al., 2026a; Ou et al., 2026).

**Sampler-learner mismatch.** Even with an accurate likelihood approximation, a more subtle issue persists. In practice, rollout samples are typically produced by *confidence-based* generation $o \sim \pi_\theta^{\text{conf}}(o \mid q)$ to navigate the combinatorial space. However, the ELBO objective still targets the likelihood of the *original* model distribution $\pi_\theta(o \mid q)$, rather than that of the actual sampling policy $\pi_\theta^{\text{conf}}(o \mid q)$, leading to a mismatch between rollout and optimization that can degrade performance (Schulman et al., 2015).

## 5.2. JustGRPO

We propose a return to simplicity. Since pure autoregressive order exhibits higher reasoning potential in our analysis (Section 4), we explicitly forgo arbitrary-order generation during the RL stage. This constraint transforms the dLLM from a chaotic sequence denoiser into a well-defined autoregressive policy.

**Formulation.** Standard GRPO assumes a policy $\pi(o_k|o_{<k}, q)$ accepting a *partial* sequence $o_{<k}$ with all tokens observed and predicts *one* token $o_k$ at a time, where $q$ is the query. Diffusion language models, however, are architected as sequence-level denoisers that accept a *full* sequence with mixed observed and masked tokens and predict the original values for *all* masked positions simultaneously.

By forgoing arbitrary-order generation, we are able to bridge the above gap and rigorously define an AR policy $\pi_\theta^{\text{AR}}$ for dLLMs. To obtain the probability of the next token $o_k$ given history $o_{<k}$, we construct $\tilde{x}_k$ where the past is observed and the future is masked:

$$\tilde{x}_k = [\underbrace{o_1, \ldots, o_{k-1}}_{\text{Observed}}, \underbrace{[\text{MASK}], \ldots, [\text{MASK}]}_{\text{Masked}}]. \quad (5)$$

Although the diffusion language model outputs predictions for all masked positions, the autoregressive policy concerns only the next token $o_k$. We define $\pi_\theta^{\text{AR}}(\cdot|o_{<k}, q)$ as follows:

$$\pi_\theta^{\text{AR}}(\cdot|o_{<k}, q) \triangleq \text{Softmax}(f_{\theta,k}(\tilde{x}_k, q)), \quad (6)$$

where $f_{\theta,k}$ denotes the model logits at position $k$. By defining a surrogate autoregressive policy $\pi_\theta^{AR}$ on top of the dLLM backbone, we convert the intractable marginalization over permutations into an exactly computable likelihood:

$$\pi_\theta^{\text{AR}}(o|q) = \prod_{k=1}^{|o|} \pi_\theta^{\text{AR}}(o_k|o_{<k}, q). \quad (7)$$

**Optimization.** The above formulation enables the direct application of standard GRPO to diffusion language models.

For each query $q$, we sample a group of outputs $\{o_i\}_{i=1}^G$ using the old policy $\pi_{\theta_{\text{old}}}^{\text{AR}}$. The objective is:

$$\mathcal{J}(\theta) = \mathbb{E}_{q \sim P(Q), \{o_i\}_{i=1}^G \sim \pi_{\theta_{\text{old}}}^{\text{AR}}} \left[ \frac{1}{G} \sum_{i=1}^G \frac{1}{|o_i|} \sum_{k=1}^{|o_i|} \right.$$

$$\left. \left( \min \left( \rho_{i,k} \hat{A}_{i,k}, \text{clip}(\rho_{i,k}, 1-\varepsilon, 1+\varepsilon) \hat{A}_{i,k} \right) - \beta \mathbb{D}_{\text{KL}} \right) \right], \quad (8)$$

where $\rho_{i,k} = \frac{\pi_\theta^{\text{AR}}(o_{i,k}|o_{i,<k}, q)}{\pi_{\theta_{\text{old}}}^{\text{AR}}(o_{i,k}|o_{i,<k}, q)}$ is the probability ratio between the current and old policies and $\hat{A}_{i,k}$ denotes the group-standardized advantage.

**Remarks.** One might worry whether training in AR mode fundamentally turns the diffusion model into a standard autoregressive model. This is not the case. The AR constraint is applied *exclusively during RL training* for more effective exploration and credit assignment. Crucially, this refines the model distribution without imposing structural constraints, *e.g.*, causal masking, that could alter the underlying structure or compromise the parallel sampling native to dLLMs. Empirically, the model remains compatible with parallel samplers (Ben-Hamu et al., 2025) to accelerate decoding at inference. JustGRPO thus benefits from AR-based reasoning exploration while preserving the parallel capability of dLLMs (see Section 6.2).

# 6. Experiments

We evaluate JustGRPO on standard mathematical reasoning and coding benchmarks. Our experimental design aims to examine two hypotheses: (i) that enforcing autoregressive (AR) order during RL training can effectively elicit reasoning capabilities without resorting to complex arbitrary-order approximations, and (ii) that this constraint applies only to the optimization objective, leaving the diffusion model's parallel decoding capabilities intact at inference.

**Experimental Setups.** We apply JustGRPO on LLaDA-Instruct (Nie et al., 2025) and evaluate its effectiveness on four standard reasoning and coding benchmarks: GSM8K, MATH-500, HumanEval, and MBPP. For mathematical tasks, we train on the official training split of each dataset following (Zhao et al., 2025; Ou et al., 2026). For coding tasks, we train on a subset of AceCoder-87K (Zeng et al., 2025) following (Gong et al., 2026; Ou et al., 2026). We apply JustGRPO directly to the models without additional task-specific SFT, using full-parameter fine-tuning. Our training recipe largely follows (Huang et al., 2025b). To evaluate the trained models, we follow the standard LLaDA evaluation protocol (Nie et al., 2025), which applies low-confidence remasking together with semi-autoregressive decoding in blocks of 32 tokens (unmasking one token per

*Table 1.* **System-level comparison** of RL approaches on LLaDA-Instruct. JustGRPO achieves strong performance despite its simplicity. Baseline numbers are mostly quoted from their original papers[2]. We note that the experimental settings are not fully consistent across these baselines. JustGRPO uses full fine-tuning and unmasks one token per step for all tasks. Among the baselines, LLaDA-1.5, LLaDOU, and ESPO also unmask one token per step, whereas the others unmask two tokens per step; LLaDA-1.5 and LLaDOU likewise use full fine-tuning, ESPO does so only on code tasks (and LoRA otherwise), and the others use LoRA. Given this heterogeneity, we additionally reproduce several representative baselines whose original settings differ from ours, under a unified protocol in Table 2, where the same conclusion holds.

| Model / Seq Len | GSM8K | | | MATH-500 | | | HumanEval | | | MBPP | | |
|---|---|---|---|---|---|---|---|---|---|---|---|---|
| | 128 | 256 | 512 | 128 | 256 | 512 | 128 | 256 | 512 | 128 | 256 | 512 |
| d1 (Zhao et al., 2025) | 73.2 | 81.1 | 82.1 | 33.8 | 38.6 | 40.2 | - | - | - | - | - | - |
| LLaDOU† (Huang et al., 2025b) | - | 88.1 | - | - | 41.1 | - | - | **59.1** | - | - | 51.6 | - |
| LLaDA-1.5‡ (Zhu et al., 2025) | - | 83.3 | - | - | - | - | 29.3 | 39.6 | **51.9** | 39.6 | 39.9 | 38.8 |
| wd1 (Tang et al., 2026) | - | 80.8 | 82.3 | - | 34.4 | 39.0 | - | - | - | - | - | - |
| *d*-TreeRPO (Pan et al., 2026) | - | 81.2 | 82.6 | - | 37.7 | 38.9 | - | - | - | - | - | - |
| ESPO (Ou et al., 2026) | 80.0 | 82.3 | 83.7 | 36.0 | 39.0 | 43.4 | 28.1 | 42.1 | 50.0 | 47.4 | 44.6 | 44.2 |
| GDPO (Rojas et al., 2026) | 78.4 | 82.8 | 84.5 | 33.2 | 39.6 | 41.4 | 26.2 | 39.6 | 39.0 | 43.6 | 50.6 | 47.1 |
| SPG (Wang et al., 2026a) | 78.5 | 86.1 | 84.5 | 33.4 | 40.0 | 41.8 | - | - | - | - | - | - |
| **JustGRPO** | **83.8** | **89.1** | **89.8** | **39.0** | **45.1** | **45.2** | **37.8** | **49.4** | **48.7** | **50.6** | **52.4** | **49.0** |

† LLaDOU modifies the base architecture with an auxiliary module.
‡ LLaDA-1.5 is trained on a privately collected dataset at a significantly larger scale.

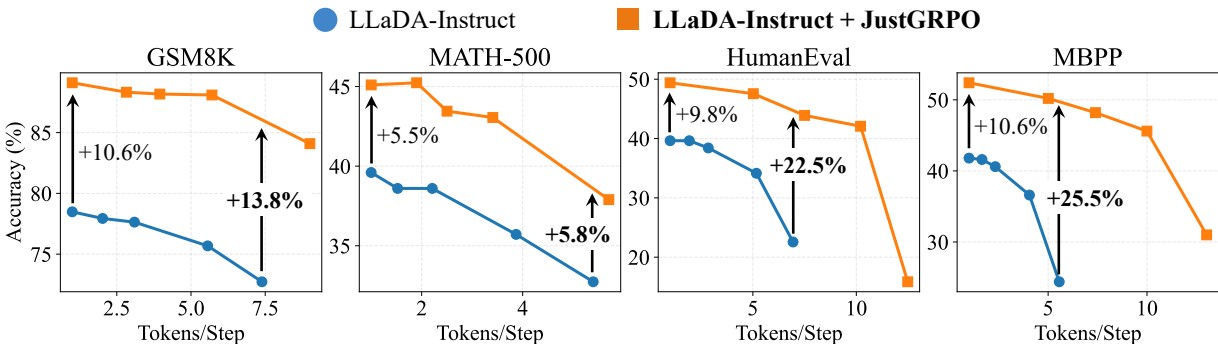

*Figure 8.* **JustGRPO preserves the parallel decoding capability of dLLMs.** Interestingly, when compared to original instruct model, accuracy gains are larger with more parallel tokens, likely due to more robust reasoning capabilities after JustGRPO training. We adopt training-free EB-sampler (Ben-Hamu et al., 2025) for parallel decoding. Generation length is set to 256.

step, *i.e.*, 256 steps for a generation length of 256), using a sampling temperature of 0. We evaluate all benchmarks at generation lengths of 128, 256, and 512 following (Zhao et al., 2025; Ou et al., 2026). More details on training are provided in Appendix A.

### 6.1. Main Results

Table 1 reports the system-level comparison. We observe that simply applying the standard GRPO formulation, despite requiring no diffusion-specific RL adaptations, already achieves strong overall performance.

**Performance on reasoning tasks.** On GSM8K, JustGRPO

---

[2]Two exceptions: the MATH-500 result of LLaDOU (41.1) is measured by us with the official checkpoint, as the original paper reports on the full MATH test set rather than MATH-500; the LLaDA-1.5 numbers are taken from (Ou et al., 2026).

*Table 2.* **Reproduction with configurations aligned to ours** (full fine-tuning, one token per decoding step, generation length 256). Reproduced baselines are marked with *. ESPO* results on HumanEval and MBPP are taken directly from the original paper, whose setting already matches ours (full fine-tuning, one token per decoding step).

| Model | GSM8K | MATH-500 | HumanEval | MBPP |
|---|---|---|---|---|
| d1* | 83.8 | 39.2 | – | – |
| ESPO* | 84.7 | 40.3 | 42.1 | 44.6 |
| SPG* | 86.9 | 41.8 | – | – |
| **JustGRPO** | **89.1** | **45.1** | **49.4** | **52.4** |

reaches 89.1% accuracy (seq len 256), competitive with or higher than prior diffusion-specific methods, with a similar trend on the more challenging MATH-500. It also remains competitive overall with LLaDOU and LLaDA-1.5, which leverage an auxiliary trainable module or larger-scale private training data and report higher accuracy on Hu-

manEval. Since the numbers in Table 1 are paper-reported under heterogeneous settings (see its caption), we additionally reproduce representative baselines (d1, ESPO, and SPG; marked with $^*$) under a unified protocol aligned with ours (full-parameter fine-tuning, one token per decoding step, generation length 256). As shown in Table 2, Just-GRPO retains clear margins (*e.g.*, +2.2/+3.3 over SPG$^*$ on GSM8K/MATH-500), indicating that optimizing over the full combinatorial space of denoising trajectories may not be necessary, and that treating the dLLM as a sequential generator during training is a simple yet effective recipe for general reasoning tasks.

### 6.2. JustGRPO Preserves Parallel Decoding

We further investigate whether the AR training compromises dLLMs' parallel decoding capabilities. We employ the training-free Entropy Bounded (EB) Sampler (Ben-Hamu et al., 2025) to evaluate inference performance under varying degrees of parallelism (tokens per step).

Figure 8 shows that our model retains full compatibility with parallel decoding and exhibits a favorable trade-off between speed and accuracy compared to the original LLaDA-Instruct model. Interestingly, the performance gain becomes more pronounced as parallelism increases. While the baseline's performance degrades sharply with high parallel steps, our model maintains relatively more stable performance. For instance, on MBPP, the accuracy gap expands from +10.6% at conservative settings (1 token/step) to +25.5% at aggressive settings ($\sim$5 tokens/step). This suggests that instead of overfitting to a specific trajectory, the AR formulation may serve as an effective training "scaffold" to refine the underlying model distribution. Such a refined distribution appears to be more resilient to the approximations inherent in parallel sampling compared to the original model.

### 6.3. Training Efficiency

Applying exact GRPO to dLLMs entails a structural trade-off: unlike autoregressive models, dLLMs must evaluate each position independently rather than in a single causal forward pass, so computing exact likelihoods incurs extra per-iteration overhead. Prior methods (Zhao et al., 2025; Ou et al., 2026) sidestep this cost by relying on approximations, which might suggest that exact optimization is impractical by comparison. Yet Figure 9 suggests otherwise: even with this overhead, JustGRPO remains competitive in its accuracy/wall-clock trade-off, matching the peak accuracy of a representative approximation-based baseline (ESPO) within comparable time.

Moreover, this overhead is far from fundamental. Its dominant source is computing the per-position probability ratios $\rho_{i,k}$ in the GRPO objective (Eq. 8). Motivated by our find-

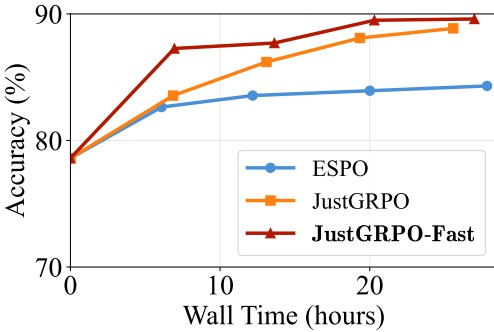

*Figure 9.* **Training efficiency on GSM8K.** Despite its larger per-iteration overhead, JustGRPO achieves a better accuracy/wall-time trade-off than the approximation-based baseline (ESPO), which saturates early. **JustGRPO-Fast**, which computes the probability ratio $\rho_{i,k}$ only at the top-25% highest-entropy positions, accelerates training further. Wall time is measured on 16×H100 GPUs.

ing that reasoning is steered by a sparse set of high-entropy forking tokens (Section 4), we introduce **JustGRPO-Fast**, which evaluates $\rho_{i,k}$ only at the *top-25% highest-entropy positions*, eliminating 75% of these forward evaluations.

Importantly, since per-token entropy can be directly recorded during the rollout stage, identifying the high-entropy positions incurs negligible overhead. Empirically, JustGRPO-Fast improves the accuracy/wall-time trade-off even over the default JustGRPO, which itself already compares favorably with the approximation-based baseline (ESPO), reaching comparable accuracy in less wall-clock time.

## 7. Conclusion

The intuitive appeal of diffusion language models lies in their order arbitrariness, often perceived as a promising mechanism for navigating complex reasoning paths. However, our study suggests that for general reasoning tasks such as mathematics and coding, this unrestricted flexibility may inadvertently constrain the reasoning potential. Specifically, arbitrary-order generation appears to prioritize the refinement of a single trajectory over broader solution coverage.

This observation points towards a simpler solution for eliciting reasoning capabilities of dLLMs. By aligning the model using a standard autoregressive objective, we enable the direct application of Group Relative Policy Optimization (GRPO) without complex adaptations tailored for order arbitrariness. This approach, JustGRPO, suggests that intentionally constraining the exploration order during training, despite its simplicity, can yield notable improvements in reasoning performance, while fully preserving the parallel decoding capabilities of dLLMs at inference. By returning to the basic left-to-right order for alignment, this work invites a reconsideration of the trade-offs between arbitrary *vs.* AR order in the development of diffusion language models.

## Impact Statement

This paper presents work whose goal is to advance the field of Machine Learning, specifically in the area of diffusion language models and reinforcement learning for reasoning tasks. Our findings suggest that simpler training approaches can be surprisingly effective, which may reduce computational costs and make these techniques more accessible. There are many potential societal consequences of our work, none of which we feel must be specifically highlighted here, as the primary contribution is methodological and aimed at improving the understanding and training of language models.

## Acknowledgements

This work is supported in part by the National Key R&D Program of China under Grant 2024YFB4708200, the National Natural Science Foundation of China under Grants U24B20173, U2541227 and 625B2107, and the Scientific Research Innovation Capability Support Project for Young Faculty under Grant ZYGXQNJSKYCXNLZCXM-I20.

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

# Appendix

## A. Experimental Details

### A.1. Data Preparation

For mathematical reasoning tasks, we train on the official training split of each dataset, following the standard protocol in (Zhao et al., 2025; Ou et al., 2026). For code generation tasks, we adopt the AceCoder-87K dataset (Zeng et al., 2025). Following the data processing pipeline of DiffuCoder (Gong et al., 2026), we further select 21K challenging samples from AceCoder-87K that are equipped with verifiable unit tests.

### A.2. Training Configuration

Our training setup largely follows (Huang et al., 2025b). Unlike (Huang et al., 2025b), we perform reinforcement learning directly on the dLLM without introducing additional trainable modules. During the rollout phase, we adopt exact autoregressive sampling, which allows direct log-probability computation under the standard GRPO formulation. We also reduce the total number of training steps, as JustGRPO exhibits fast and stable convergence. All experiments are conducted on $16\times$ NVIDIA H100 GPUs. Detailed hyperparameters are reported in Table 3.

*Table 3.* **Training hyperparameters** for JustGRPO[3].

| Hyperparameter | Value |
|---|---|
| Base Model | LLaDA 8B Instruct |
| RL Algorithm | GRPO |
| Optimizer | AdamW |
| Learning Rate | $5 \times 10^{-6}$ |
| LR Scheduler | Constant |
| Weight Decay | 0.0 |
| Optimizer Betas $(\beta_1, \beta_2)$ | $(0.9, 0.999)$ |
| Global Batch Size | 64 |
| Group Size $(G)$ | 16 |
| Policy Update Steps | 1 |
| Training Steps | 125 |
| Max Completion Length | 256 |
| Sampling Temperature | 1.0 |
| KL Penalty Coefficient | 0.0 |

### A.3. Reward Function

**Mathematical Reasoning Tasks.** We employ a binary reward scheme. Each completion receives a reward of 1 if and only if the final answer is mathematically equivalent to the ground-truth solution following (Huang et al., 2025b).

**Code Generation Tasks.** For code generation, the total reward $r$ is defined as the sum of a correctness reward $r_{\text{code}}$ and a format reward $r_{\text{format}}$:

$$r = r_{\text{code}} + r_{\text{format}}. \quad (9)$$

- **Correctness reward ($r_{\text{code}}$):** Defined as the pass rate (ranging from 0 to 1) of the generated code on the provided unit tests. This term is only evaluated when $r_{\text{format}} = 1$.

- **Format reward ($r_{\text{format}}$):** A heuristic reward designed to encourage syntactically valid outputs.

  - $1.0$: A valid Markdown code block with correct Python syntax.
  - $0.5$: A valid Markdown code block that contains syntax errors.
  - $0.0$: Failure to generate a valid Markdown code block.

## B. More Analysis Results on Reasoning Boundary

To further validate the robustness of our findings about reasoning boundary in Section 4, we conduct extended analyses on the HumanEval benchmark using the LLaDA-Instruct model (Nie et al., 2025). We investigate the impact of temperature and sampling strategies on reasoning performance.

### B.1. Temperature Analysis

As shown in Figure 10, the AR mode exhibits a standard pattern: performance peaks at moderate temperatures ($T \approx 0.6$) and degrades when $T > 1.0$, whereas the Arbitrary Order mode attains its peak performance at higher temperatures.

This observation aligns with the *entropy degradation* mechanism discussed in Section 4: the diffusion sampler inherently suppresses uncertainty at critical branching points, thereby requiring higher temperatures to induce sufficient exploration. Notably, even under these optimized settings, the arbitrary order mode does not match the reasoning potential of the AR mode in our experiments. As illustrated in the "Optimal" comparison setting (bottom right of Figure 10), the best-performing AR configuration still shows better scaling behavior than the optimal arbitrary order baseline. A plausible explanation is that excessively high temperatures in arbitrary order decoding inject noise into tokens that require high determinism (*e.g.*, code syntax or mathematical suffixes), leading to nonsensical outputs and degraded results (Wang et al., 2025; Huang et al., 2025a).

---

[3]Empirically, GSM8K converges much faster (around 50 steps already reaches ~89% accuracy), while coding tasks (HumanEval, MBPP) are slower and benefit from the full 125 steps. We therefore used 125 steps uniformly across all datasets for consistency.

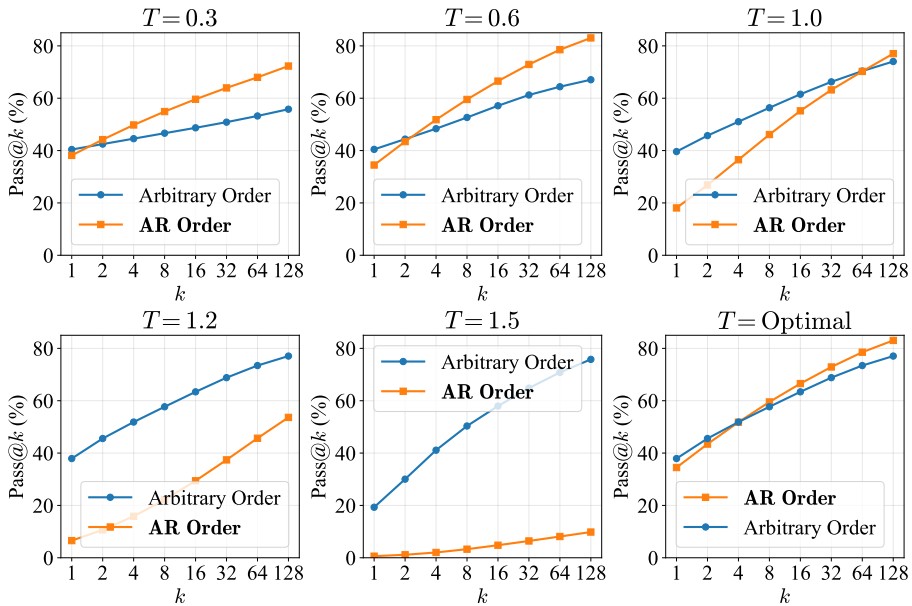

*Figure 10.* **Pass@$k$ comparison with different temperatures**.

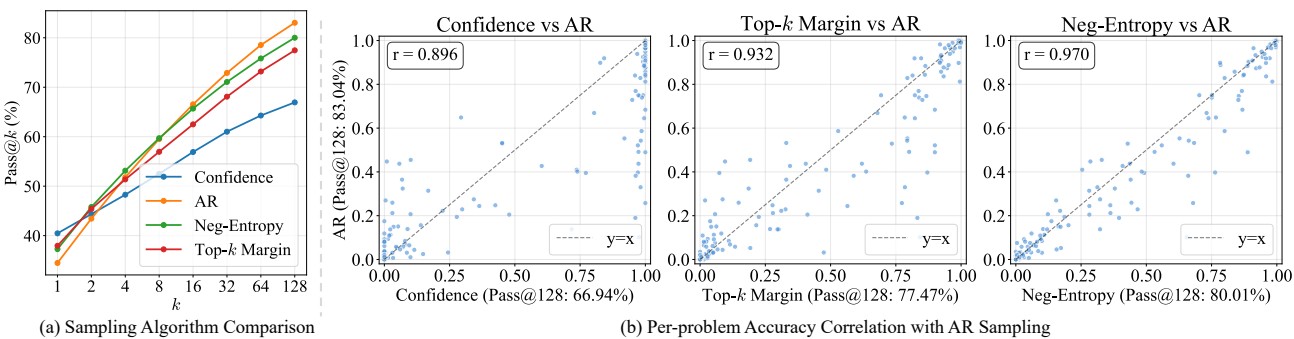

*Figure 11.* **(a) Sampling algorithm comparison**. **(b) Correlation between different sampling algorithms and AR in per-problem accuracy**. Algorithms with higher Pass@$k$ performance tend to show *higher correlation with AR* in problem-level performance.

## B.2. Different Sampling Algorithms

We also experiment with different sampling algorithms, such as negative entropy sampling (Neg-Entropy) (Ye et al., 2025b) and top-$k$ margin sampling (Margin) (Kim et al., 2025) in Figure 11(a). The results show that although more sophisticated sampling algorithms can achieve better Pass@$k$ performance than default confidence-based sampling, they still cannot catch up with AR order. Meanwhile, these better Pass@$k$ algorithms also show slightly worse Pass@1 performance compared to confidence-based sampling, making the overall Pass@$k$ curve closer to AR order's Pass@$k$ curve. To investigate this similarity, we calculate the per-problem accuracy correlation between different sampling algorithms and the AR mode (Figure 11(b)). We observe that algorithms with higher scaling potential (higher Pass@128) consistently show stronger correlation with AR, with the most effective method (Neg-Entropy) achieving

a correlation of 0.970. This suggests that sampling algorithms with better Pass@$k$ tend to behave more like AR in performance characteristics.

## B.3. More Results on Entropy Degradation

To validate the robustness of the "entropy degradation" phenomenon observed in Section 4, we extended our analysis to a wider range of logical connectors. We conducted experiments on a comprehensive set of common connectives that typically serve as forking tokens in reasoning chains.

The results, shown in Figure 12, show that the phenomenon is consistent across these diverse tokens. Similar to the primary findings, the AR order maintains higher average entropy at these forks, indicating active reasoning and decision-making. Conversely, the Arbitrary Order consistently results in lower entropy, consistent with the premature collapse of branching possibilities.

The specific forking tokens evaluated in this extended experiment include: "Therefore", "Thus", "So", "Since", "When", "Given", "However", "Let", "First", "Then", "Next", "Finally", "Now", "Similarly", "Calculate", "Solving", "Notice", "Specifically", "Follows", "Because", "But", "Or", "Consider", "Also", "Express", and "Write".

## C. Is Random Order an Alternative?

Our main analysis in Section 4 concerns how dLLMs are used in practice, where decoding is driven by confidence-based sampling and semi-autoregressive sampling (Nie et al., 2025; Zhao et al., 2025). This is the setting in which bypassing occurs and entropy degrades, and it is the setting our analysis targets.

A dLLM, however, is not required to use such a sampler. A natural question is whether a random decoding order, which connects to the original dLLM formulation (Sahoo et al., 2024; Shi et al., 2024), would behave differently. We find that it does not help, either.

**Random order does not help coverage, and it breaks generation.** Under Pass@128, random order is at best comparable to confidence-based sampling, and it still falls well short of AR order (Table 4, top). Its Pass@1 is far worse, well below both baselines (Table 4, bottom). The reason is straightforward. In a dLLM, each token is predicted from its surrounding context, but a random order often requires the model to decide a token before that context exists, for example predicting token 30 when only token 1 is known. The resulting output is not merely incorrect, but structurally broken, as the following LLaDA rollout illustrates:

```
3.  **Third Month:**
- The number of downloads reduced by
30% compared to the second month.
\[ 180 - (   \times 180 - (   - 180)
= 180 - 0.30 \times 180 = 54 ...\]
```

The LaTeX is corrupted, with dangling operators and missing operands. Outputs of this kind rarely earn a reward, which limits what RL can learn from.

**Random order does not survive RL either.** We then bring random order into our own training setup. To apply an exact likelihood factorization akin to JustGRPO, we adopt a single random permutation and keep it fixed throughout training. We call this variant JustGRPO-Random. Keeping the rest of the pipeline unchanged, JustGRPO-Random only reaches 82.2% on GSM8K, compared to 89.1% for AR order (Table 5). A random order thus does not appear to be a remedy: in our experiments, it neither widens coverage as a sampler nor delivers strong reasoning after RL.

*Table 4.* **Random order does not improve coverage and collapses single-shot accuracy.** Pass@128 (top) and Pass@1 (bottom) for confidence-based arbitrary order, fully random order, and AR order, measured on LLaDA-Instruct.

| Method | GSM8K | MATH-500 | MBPP | HumanEval |
|---|---|---|---|---|
| *Pass@128(%)* | | | | |
| Confidence-based | 97.0 | 71.4 | 67.1 | 67.1 |
| Fully Random | 97.5 | 70.6 | 71.8 | 64.9 |
| AR (Left-to-right) | **99.0** | **75.6** | **78.7** | **83.0** |
| *Pass@1(%)* | | | | |
| Confidence-based | **78.6** | **30.4** | **40.7** | **40.5** |
| Fully Random | 43.3 | 14.1 | 15.0 | 12.4 |
| AR (Left-to-right) | 78.0 | 27.9 | 34.9 | 34.5 |

*Table 5.* **Random order also fails in RL post-training.** *JustGRPO-Random* is identical to JustGRPO except that it decodes in a fixed random order rather than left-to-right. Results are measured on GSM8K with LLaDA-Instruct.

| Method | GSM8K Acc. | Δ |
|---|---|---|
| LLaDA-Instruct | 78.6% | — |
| JustGRPO-Random | 82.2% | +3.6% |
| JustGRPO (AR order) | **89.1%** | +10.5% |

## D. General Capability Preservation

Beyond reasoning, we verify that JustGRPO preserves the model's general capabilities. We evaluate the previously trained JustGRPO checkpoint against the original LLaDA-Instruct model on four standard non-reasoning benchmarks: MMLU, MMLU-Pro, HellaSwag, and ARC-C. As reported in Table 6, general capabilities are largely preserved, with only small task-dependent fluctuations. This is expected: the AR formulation in JustGRPO acts purely as a training-time scaffold for RL exploration rather than a structural constraint on the model. The architecture, attention pattern, and pretrained knowledge are largely preserved, so JustGRPO improves how the model explores reasoning trajectories during RL without interfering with how it encodes and expresses general knowledge.

*Table 6.* **General (non-reasoning) capabilities are preserved** after JustGRPO. Accuracy (%) on four standard benchmarks for the base model and the JustGRPO checkpoint.

| Model | MMLU | MMLU-Pro | HellaSwag | ARC-C |
|---|---|---|---|---|
| LLaDA-Instruct | 65.5 | **37.0** | 74.6 | **88.5** |
| JustGRPO | **65.8** | 36.7 | **74.8** | 87.5 |

**Note on bidirectional refinement.** Because JustGRPO imposes the AR factorization only as a training-time scaffold rather than a constraint on inference, we expect it to largely preserve the bidirectional behavior characteristic of dLLMs. This expectation is consistent with the preserved parallel decoding capability reported in Section 6.2. A natural extension is to ask whether dLLMs can iteratively refine and revise their already-decoded outputs. This connects to cor-

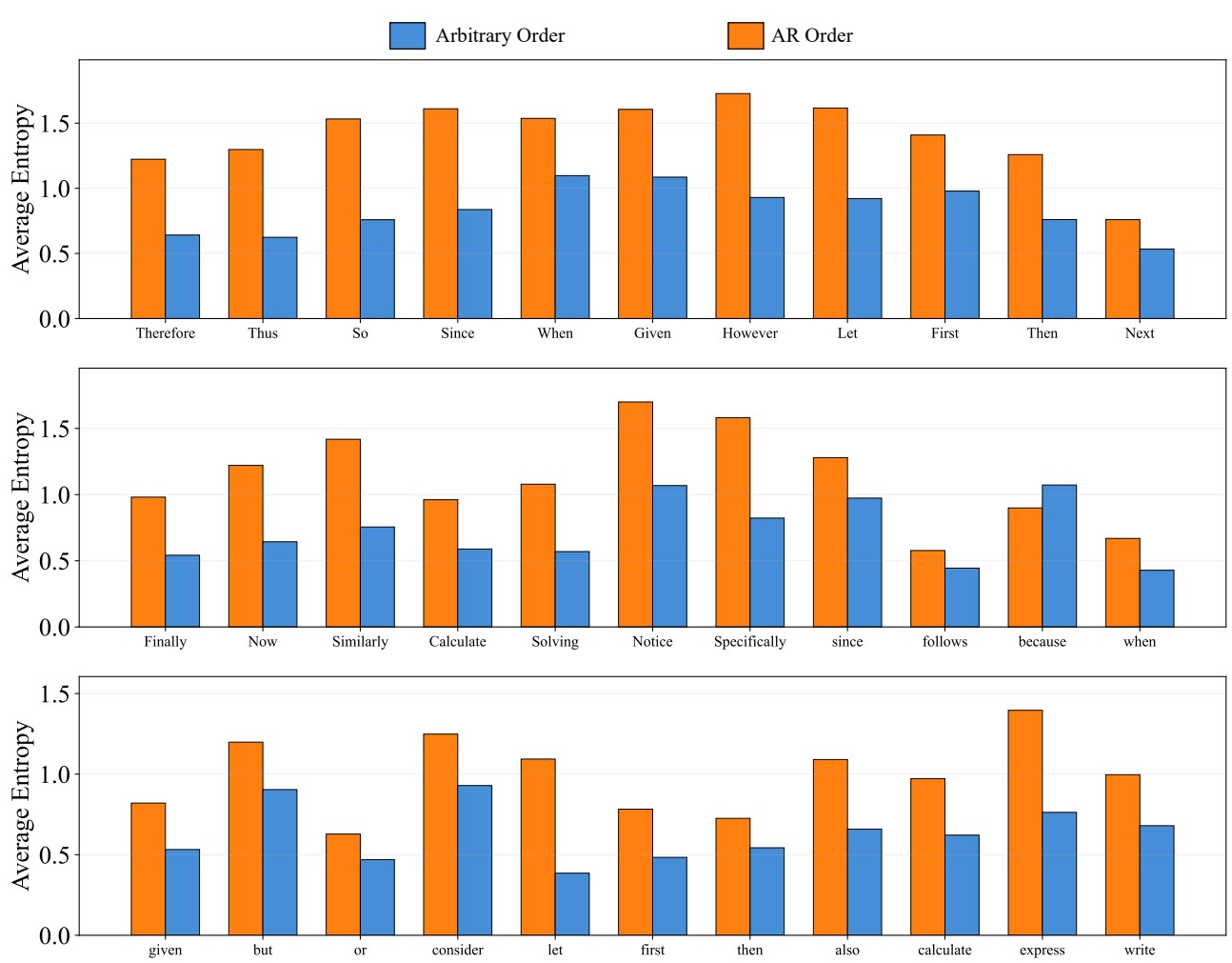

*Figure 12.* **Entropy comparison results on more forking tokens**.

rective approaches such as CDLM (Zhang et al., 2025b) and to the bidirectional-context settings studied in Parallel-Bench (Kang et al., 2026), and we view it as a promising direction that could complement JustGRPO.

