# OpenReview forum: "The Flexibility Trap: Rethinking the Value of Arbitrary Order in Diffusion Language Models"
_ICML.cc/2026/Conference — ICML 2026 spotlight_

### Official Review · Reviewer_ps3v · 2026-03-06

**Soundness:** 2
**Presentation:** 3
**Significance:** 2
**Originality:** 3
**Overall Recommendation:** 5
**Confidence:** 4

**Summary:**

This paper introduces JustGRPO, a reinforcement learning framework that resolves the reasoning deficiency in Diffusion Language Models (dLLMs) by addressing the "Flexibility Trap," where arbitrary-order generation allows models to bypass critical logical forks during training. By imposing a strict autoregressive (AR) scaffold during the reinforcement learning phase, the authors force the model to confront every causal dependency in sequence, enabling exact likelihood estimation and more effective policy gradient updates. The resulting model achieves state-of-the-art performance on benchmarks like GSM8K and MATH-500, proving that the "logic-blindness" of diffusion models can be mitigated through a structured training objective.

**Compliance With Llm Reviewing Policy:**

Affirmed.

**Final Justification:**

The rebuttal adressed my concerns.

**Key Questions For Authors:**

I am curious about the impact of the unidirectional RL signal training on Bidirectional Refinement and Remasking:

JustGRPO treats the dLLM as a strictly autoregressive policy during RL training, masking all "future" tokens to compute exact likelihoods. While this improves sequential reasoning, it potentially biases the model to rely solely on past context, neglecting the bidirectional attention that is the hallmark of diffusion architectures.

Question: Have the authors conducted evaluations to determine if this unidirectional RL signal impacts the model's performance on tasks specifically requiring iterative error correction? If such data is not yet available, would the authors consider adopting an experimental framework similar to ParallelBench (as seen in arXiv:2510.04767 and arXiv:2512.15596) to verify if the RL-induced focus on past context affects the model’s signature ability to refine previous outputs based on future information?

**Limitations:**

yes

**Strengths And Weaknesses:**

Strengths:
1. Insightful Identification of the "Flexibility Trap": The paper provides a compelling analysis of how arbitrary-order generation allows models to bypass high-uncertainty "logical forks," such as connectives like "Therefore" or "Since". This phenomenon, termed "entropy degradation," reveals that the model effectively prunes its own reasoning space by resolving easy tokens first, thereby limiting its ability to explore diverse and correct solution paths during reinforcement learning.
2. Methodological Simplicity: By introducing JustGRPO, the authors offer a minimalist yet effective solution that treats dLLMs as autoregressive policies during training to avoid the combinatorial explosion of $O(N!)$ denoising trajectories. This approach not only achieves state-of-the-art results on benchmarks but also successfully decouples training exploration from inference execution, fully preserving the parallel decoding efficiency native to diffusion models.


Weakness:

1. While the effectiveness of JustGRPO is compelling, the study could more thoroughly decouple the mathematical advantages of exact likelihood estimation from the logical benefits of left-to-right causal directionality. The current comparison between the adaptive Arbitrary Order (AO) baseline and the strict Autoregressive (AR) order involves two simultaneous shifts which include the precision of the optimization objective and the sequence of token resolution. Because the AO baseline relies on approximate likelihoods and permits the bypassing of logical forks, it remains difficult to determine if JustGRPO succeeds primarily due to its superior mathematical framework or its alignment with the step-by-step nature of reasoning. To resolve this ambiguity, providing a Fixed Random Order baseline where tokens are decoded one by one but non-causal sequence would offer more clarity. This baseline would maintain the same mathematical advantages as JustGRPO, specifically the exact likelihood calculation and the prevention of logical fork bypassing, while isolating the effect of directionality. If JustGRPO significantly outperforms such a fixed random baseline, it would provide much stronger empirical support for the thesis that left-to-right directionality is a fundamental requirement for eliciting reasoning potential in diffusion language models rather than just a computational convenience.

---

> ### Author Rebuttal · Authors · 2026-03-31
>
> Many thanks for your careful and constructive review. We appreciate your recognition of our work and your helpful suggestions. We address your points below.
>
> ---
>
> **[W1] *Disentangling Exact Likelihood from Left-to-Right Directionality***
>
> Thanks for this valuable advice. We trained a Fixed Random Order baseline where samples share a randomly permuted order, with tokens decoded one by one.
>
> **Ablation result.** This ablation helps disentangle the two effects. Replacing approximate surrogates with exact token-level optimization (Random vs. base) yields a meaningful but modest +3.6%. Left-to-right directionality (AR vs. Random) contributes an additional +6.9%, roughly **2x the gain** compared to previous. This suggests that *left-to-right directionality is the more dominant factor*.
>
> | Method | GSM8K Accuracy | Δ |
> | --- | --- | --- |
> | LLaDA-Instruct | 78.6% | — |
> | JustGRPO-Random | 82.2% | +3.6% |
> | JustGRPO (AR order) | **89.1%** | **+10.5%** |
>
> **The intuition.** Indeed, fixed random order maintain similar mathematical advantages of JustGRPO, but it may force the model to make branching decisions before the most relevant prior reasoning context has been sufficiently established. In contrast, left-to-right order presents each potential fork after the necessary preceding context is available, while still withholding future tokens that would prematurely collapse alternatives. We believe this is why AR order yields more effective exploration for reasoning.
>
> We will incorporate this ablation into the revised manuscript.
>
> ---
>
> **[Q1] *Impact of Unidirectional RL on Bidirectional Refinement***
>
> We thank the reviewer for this thoughtful question. We address it with both direct evaluation and further analysis.
>
> **ParallelBench evaluation.** Following the reviewer’s suggestion, we evaluate on the Waiting Line task from ParallelBench, which stress-tests the model’s ability to utilize bidirectional context.
>
> | **Task** | **LLaDA** | **LLaDA + JustGRPO** |
> | --- | --- | --- |
> | Copy | 100 | 100 |
> | Sort | 0 | 0 |
> | Reverse | 97 | 97 |
> | Shuffle | 21 | 20 |
> | Insert Index | 14 | 12 |
> | Insert Random | 85 | 89 |
> | Remove Index | 18 | 23 |
> | Remove Random | 82 | 91 |
> | Replace Index | 15 | 13 |
> | Replace Random | 58 | 60 |
>
> > We follow the evaluation protocol from Tab. 5 of the ParallelBench paper, with LLaDA results also taken from Tab. 5.
>
> We observe **no systematic degradation after JustGRPO**: performance is unchanged on Copy/Reverse, marginally lower on a few tasks (e.g., Shuffle: 21 to 20), and slightly improved on several other tasks such as Insert Random (85 to 89). This is consistent with the fact that JustGRPO only imposes an AR factorization during RL training, without modifying the bidirectional attention of dLLMs.
>
> **On iterative error correction.** To probe if unidirectional RL signal affects the model's capacity for iterative refinement, we preliminarily experimented with a remasking-based sampler (ReMDM), which allows previously decoded tokens to be re-masked and regenerated. We compare LLaDA and LLaDA+JustGRPO on the Replace Index task with and without remasking:
>
> | **Task** | **LLaDA** | **LLaDA + JustGRPO** |
> | --- | --- | --- |
> | without ReMDM | 15 | 13 |
> | with ReMDM | 14 | 13 |
>
> Remasking provides limited improvement for **both** the base model and the RL-trained model. This aligns with findings in ParallelBench (Figures 18, 19) and the analysis in CDLM (2512.15596): current dLLMs lack the error-aware confidence needed to identify which tokens should be re-masked, making training-free remasking samplers largely ineffective. **JustGRPO neither worsens nor changes this behavior**, though we acknowledge the training-free sampler experiment may not be the most sensitive probe for refinement capacity.
>
> We believe that dedicated refinement-aware training objectives, such as those proposed in CDLM, offer a more principled path to unlocking iterative correction in dLLMs, and we are actively exploring combining such objectives with JustGRPO.
>
> We will discuss and cite both ParallelBench and CDLM in the revised manuscript. We thank the reviewer again for highlighting this important direction.

---

> > ### Author Rebuttal · Reviewer_ps3v · 2026-04-03
> >
> > Thank you for the thorough rebuttal. The Fixed Random Order ablation convincingly disentangles the two confounded factors, and I appreciate the commitment to including it in the revision. The ParallelBench evaluation is also encouraging and partially addresses our concern. However, I note that the current comparison only includes LLaDA (base) vs. LLaDA + JustGRPO; adding LLaDA + Vanilla RL (d1 / ESPO / SPG) as a baseline would better isolate whether the performance is due to finetuning or specific to JustGRPO's AR training protocol. I consider W1 substantively addressed and Q1 partially addressed, and am willing to raise my score upon inclusion of the suggested baseline.

---

> > > ### Author Response · Authors · 2026-04-06
> > >
> > > Thank you for the specific and actionable suggestion. We address it below.
> > >
> > > **ParallelBench results with ESPO baseline.** We trained LLaDA + ESPO (a representative non-AR RL method) under the same base model and RL budget as JustGRPO. The key difference is that ESPO follows the standard non-AR, approximation-based objective, while JustGRPO imposes the AR scaffold. This lets us test whether the preserved bidirectional behavior is simply a generic consequence of RL fine-tuning, or whether the choice of training protocol plays a meaningful role.
> > >
> > > | **Task** | **LLaDA** | **LLaDA + ESPO** | **LLaDA + JustGRPO** |
> > > | --- | --- | --- | --- |
> > > | Copy | 100 | 100 | 100 |
> > > | Sort | 0 | 0 | 0 |
> > > | Reverse | 97 | 91 | 97 |
> > > | Shuffle | 21 | 12 | 20 |
> > > | Insert Index | 14 | 19 | 12 |
> > > | Insert Random | 85 | 85 | 89 |
> > > | Remove Index | 18 | 20 | 23 |
> > > | Remove Random | 82 | 66 | 91 |
> > > | Replace Index | 15 | 12 | 13 |
> > > | Replace Random | 58 | 67 | 60 |
> > > | Average | 49.0 | 47.2 | 50.5 |
> > >
> > > The results indicate that preserving bidirectional behavior is **not** merely a generic consequence of RL fine-tuning. Specifically, we observe that while ESPO shows gains on a few tasks (Insert Index, Replace Random), it exhibits notable degradation on several bidirectional tasks (Remove Random: 82→66; Shuffle: 21→12), leading to a drop in average score (49.0→47.2). In contrast, JustGRPO stays much closer to the base model's performance and even slightly improves upon it overall (49.0→50.5).
> > >
> > > **Remasking probe.** For completeness, we added ESPO to the ReMDM comparison on Replace Index:
> > >
> > > | **Replace Index** | **LLaDA** | **LLaDA + ESPO** | **LLaDA + JustGRPO** |
> > > | --- | --- | --- | --- |
> > > | without ReMDM | 15 | 12 | 13 |
> > > | with ReMDM | 14 | 13 | 13 |
> > >
> > > All three models show similarly limited benefit from remasking, indicating this limitation is more likely inherent to the base model than introduced by either RL method.
> > >
> > > We hope this additional baseline more adequately addresses the reviewer's remaining concern regarding Q1 and thank the reviewer again for the suggestion, which has meaningfully strengthened the empirical narrative of our work.

---

### Official Review · Reviewer_BMXv · 2026-03-11

**Soundness:** 3
**Presentation:** 3
**Significance:** 3
**Originality:** 4
**Overall Recommendation:** 5
**Confidence:** 5

**Summary:**

This paper addresses the missing conception of the diffusion LLM + RL methods; the training rollout via diffusion-style generation exhibits `the flexibility trap'. Two main issues (taxes) of those are (1) likelihood intractability issue and (2) entropy degradation. Per this finding, they propose 'Just GRPO', for which training rollout / likelihood computation both use the strict causal ordering. Just GRPO is handy to implement and claimed to have superior performance compared to baselines.

**Compliance With Llm Reviewing Policy:**

Affirmed.

**Final Justification:**

justified in the response. (I appreciate the idea of the fast just-GRPO, whose details are clarified in the second rebuttal phase). I believe this paper not only throws a strong message to the dLLM + RL community but also is now armed with a thorough evaluation / empirical claim.

**Key Questions For Authors:**

Please refer to the 'weakness' section.

**Limitations:**

Please refer to the 'weakness' section.

**Strengths And Weaknesses:**

**Strengths.**
The paper is very well written and clearly articulates its central claim, supported by experimental evidence. In particular, the authors identify two key issues in existing diffusion LLM + RL approaches and provide a thoughtful analysis of these limitations. I believe these insights could be valuable for designing more efficient and principled dLLM + RL methods.

**Weaknesses.**
However, there are several important experimental details that are not clearly specified, which makes it difficult to assess whether the comparisons with the baselines are fully fair.

1. It is unclear whether the proposed method uses **full fine-tuning** or **parameter-efficient fine-tuning (e.g., LoRA)**.
2. The **number of decoding steps used during evaluation** is not clearly stated.

As far as I understand, the baseline results reported in Table 1 appear to be taken directly from prior work. Those baselines typically use **LoRA fine-tuning and 128 decoding steps during evaluation**. To ensure a fair comparison, the evaluation protocol and PEFT setup should ideally be consistent across all methods, or the baselines should be re-implemented under the same configuration. It would be helpful if the authors could clarify these details and indicate whether a **reproducible codebase** with the exact training and evaluation settings will be released.

I also have one technical question. Unlike vanilla GRPO for dLLMs, where the likelihood can be evaluated in a single pass, my understanding is that **JustGRPO requires separate forward passes for each causal-like sequence when computing the likelihood** (please correct me if I am mistaken). If so, this could introduce additional computational overhead. More generally, it seems that this limitation may still persist due to the any-order generation nature of dLLMs.

Overall, I find the paper’s position and motivation novel and well supported conceptually. However, several experimental details would benefit from further clarification. If the authors can clearly address these points, I would be happy to increase my score (e.g., to 5 or 6).

---

> ### Author Rebuttal · Authors · 2026-03-31
>
> Thank you for recognizing the novelty of our work and your careful, professional review, which are very helpful for improving our paper. We address each point below.
>
> ---
>
> **[W1]: *Experimental Fairness (Fine-tuning Strategy, Decoding Steps, Reproducibility)***
>
> Thanks for raising this important point. We are happy to clarify and provide additional controlled evidence.
>
> JustGRPO uses *full fine-tuning and 1 token per step decoding (e.g., 256 steps for 256 generation length)* during evaluation. While some baselines (e.g., ESPO on code tasks, LLaDA-1.5 and LLaDOU) in Table 1 share these choices, we acknowledge that others differ. To ensure a controlled comparison, we re-implemented (marked in \*) representative baselines under our settings (full fine-tuning, 256 decoding steps, generation length 256).
>
> | Model | GSM8K | MATH-500 | HumanEval | MBPP |
> |---|---|---|---|---|
> | d1* | 83.8 | 39.2 | - | - |
> | ESPO* | 84.7 | 40.3 | 42.1 | 44.6 |
> | SPG* | 86.9 | 41.8 | - | - |
> | JustGRPO | **89.1** | **45.1** | **49.4** | **52.4** |
>
> > \*: our re-implemented baselines. ESPO results on HumanEval and MBPP are from the original paper as they already use full fine-tuning and 256-step decoding (*same as our protocol*).
>
> Under matched protocol, JustGRPO **still outperforms all re-implemented baselines by notable margins (e.g., +2.2/+3.3 over SPG\* on GSM8K/MATH-500)**, suggesting that *the main conclusion is robust to the fine-tuning and evaluation setup*. We will revise the manuscript to explicitly state all configurations and replace baseline results with our reproduced versions to ensure fairness.
>
> **Reproducibility commitment.** We are happy to release our complete codebase, including training scripts and evaluation settings to facilitate verification.
>
> ---
>
> **[W2]: *Computational Overhead of Per-Position Likelihood***
>
> **The reviewer's understanding is correct**: JustGRPO requires per-position forward passes for exact likelihood computation (currently described in Appendix D, which we will clarify in the main text).
> However, JustGRPO demonstrates competitive accuracy vs. wall-time tradeoff compared to baselines (GSM8K, generation length 256; wall-clock time benchmarked with 16$\times$ H100 GPUs):
>
> | Method        | Acc. at ~10h | Acc. at ~20h | Acc. at ~30h |
> | ------------- | ------------ | ------------ | ------------ |
> | ESPO*        | 84.4%       | 84.6%        | 84.7%       |
> | SPG*         | **86.5%**   | 86.8%        | 86.9%       |
> | JustGRPO      | 85.0%       | **88.1%**   | **89.1%**   |
>
> > \*: our re-implemented baselines, see response to [W1]
>
> While JustGRPO is slightly slower initially, it **surpasses all baselines after moderate training and continues to improve**, whereas baselines largely saturate. We believe this is because exact likelihood estimation provides a cleaner learning signal. In contrast, while approximation-based approaches offer faster per-iteration speed, their estimation error may produce noisier gradients that leads to earlier performance saturation, as reflected above.
>
> **Regarding the broader architectural point.** The reviewer is correct that this overhead is structural. Without causal masking, dLLMs cannot obtain all positional likelihoods via a single forward pass as AR models do. However, **this cost is highly amenable to reduction**. JustGRPO-Fast (Appendix D) restricts gradient computation to only the top-25% high-entropy tokens, motivated by our finding that reasoning is disproportionately steered by these forking tokens. As shown below, **JustGRPO-Fast surpasses all baselines at comparable time budgets (87.1% acc. at ~10h)**.
>
> | Method | Acc. at ~10h | Acc. at ~20h | Acc. at ~30h |
> |---|---|---|---|
> | Best Baseline (SPG*) | 86.5% | 86.8% | 86.9% |
> | JustGRPO | 85.0% | 88.1% | 89.1% |
> | JustGRPO-Fast | **87.1%** | **89.8%** | **89.8%** |
>
> We will add detailed discussion on computational overhead of JustGRPO in our revised manuscript.

---

> > ### Author Rebuttal · Reviewer_BMXv · 2026-04-03
> >
> > I thank the authors for the additional experiments and clarifications. Overall, my experimental concerns are largely resolved, as the rebuttal provides a much more controlled and informative set of comparisons. I strongly encourage the authors to incorporate these points clearly into the revised manuscript, as I believe they would be valuable to the broader community.
> >
> > I still have one remaining question regarding JustGRPO-Fast. My understanding was that the main source of slowdown in JustGRPO comes from requiring a separate forward pass for each position when computing the exact likelihood, rather than from computing the loss on all tokens. If so, it is still unclear to me how restricting gradient computation to the top-25% high-entropy tokens leads to such a substantial wall-clock speedup. I would appreciate further clarification on this point.
> >
> > Given the new evidence, I am raising my score from 3 to 4 and I would be happy to raise it further.

---

> > > ### Author Response · Authors · 2026-04-06
> > >
> > > Thank you for the precise follow-up. We acknowledge that our original wording was not precise enough and are happy to clarify.
> > >
> > > The speedup does **not** come from forwarding at all positions and then masking gradients on 75% of them; if that were the case, the wall-clock gain would indeed be marginal. Instead, JustGRPO-Fast **directly skips forward passes at non-selected positions**.
> > >
> > > Concretely, in the GRPO objective (Eq. (8)), the dominant cost lies in computing the per-position probability ratios $\rho_{i,k}$:
> > >
> > > - **JustGRPO** computes $\rho_{i,k}$ at **every** position $k$.
> > > - **JustGRPO-Fast** computes $\rho_{i,k}$ **only at the top-25% highest-entropy positions**, eliminating 75% of these forward passes.
> > >
> > > Since per-token entropy can be directly recorded during the rollout stage, identifying the high-entropy positions introduces negligible overhead. Combining the rollout stage (identical cost for both variants) with the accelerated policy-update stage yields a **~2.3x wall-clock speedup** per iteration for JustGRPO-Fast.
> > >
> > > Although optimizing over only the high-entropy subset makes the objective a sparse surrogate, the favorable accuracy/wall-time tradeoff reported in our earlier response suggests that these high-entropy tokens, which correspond to key reasoning decisions, carry sufficient learning signal for effective policy improvement.
> > >
> > > We hope this clarification resolves your concern, and we will revise the manuscript with a precise description to make this mechanism unambiguous.

---

### Official Review · Reviewer_eiGf · 2026-03-13

**Soundness:** 2
**Presentation:** 2
**Significance:** 3
**Originality:** 3
**Overall Recommendation:** 4
**Confidence:** 4

**Summary:**

This paper challenges the advantage of arbitrary generation order of diffusion LLM, which has been widely adopted for sampling in enhancing the reasoning of dLLMs through policy-gradient based methods such as GRPO. Specifically, this paper finds that although arbitrary order generation have higher pass@1 score than AR generation, it has lower pass@k values with higher k than AR generatoin. And pass@k is important for RL optimization since algorithms like GRPO depends on group sampling. With this oberservation, they proceed to treat dLLMs as an AR llm and sample autoregressively (1 token at a time) and compute likelihood "exactly" assuming chain-rule applied here. And this has shown to have better performance then prior GRPO methods which aims to improve the likelihood estimation's tightness.

**Compliance With Llm Reviewing Policy:**

Affirmed.

**Final Justification:**

The rebuttal adressed my concerns and I remain my score as stated in my response.

**Key Questions For Authors:**

please see aobve.

**Limitations:**

please see above.

**Strengths And Weaknesses:**

Pros:
1. This paper provides a fresh angle in how to do sampling for group-based policy gradient methods for diffusion LLM, where prior methods such as diffu-grpo, spg have adopted low-confidence remasking rules. This can be seen as a cheating mechanism for dLLM where it skips exploring high-entropy branching point, which can be very important to roll out diverse and correct answers. This paper provides valuable insight on how this greedy decoding can hurt RL sampling.
2. Very strong performance as compared to other SoTA methods for dLLM.

Cons:

1. this paper has compared against AR generation with confidence-based arbitrary sampling, and block-wise AR sampling. But it has not compared against a purely random AO baseline, which can still have possibility to sample hard branching tokens. And if this works as well, we don't need fully AR generation.
2. In Table 1, for SPG and other baselines, what sampling methods are used? Is JustGRPO requiring sampling one token at a time, without KV cache, this is very slow as compared to other sampling which might sample multi-tokens at one step. The compute used for sampling is unmatched, which might be a another source of performance gap. Would be great to see a learning efficiency comparison between the baseline such as between JustGRPO and SPG.

---

> ### Author Rebuttal · Authors · 2026-03-29
>
> We sincerely thank the reviewer for the constructive feedback and for recognizing our novel perspective. Below we address each concern.
>
> ---
>
> **[W1] *Comparison against a purely random AO baseline.***
>
> Thank you for this valuable suggestion. We believe the results below further clarify why AR order is particularly suited for reasoning.
>
> **Empirical results.** A fully random order does bypass the "easy-first" bias of confidence-based sampling, yet it does **not** translate into better solution coverage. Under Pass@128, random AO performs only comparably to confidence-based sampling, while remaining clearly behind AR order:
>
> |Method (Pass@128)|GSM8K|MATH-500|MBPP|HumanEval|
> |---|---|---|---|---|
> |Confidence-based|97.0|71.4|67.1|67.1|
> |Fully Random|97.5|70.6|71.8|64.9|
> |AR (Left-to-right)|**99.0**|**75.6**|**78.7**|**83.0**|
>
> More critically, random AO's Pass@1 drops **substantially**, far below both confidence-based and AR baselines:
>
> |Method (Pass@1)|GSM8K|MATH-500|MBPP|HumanEval|
> |---|---|---|---|---|
> |Confidence-based|**78.6**|**30.4**|**40.7**|**40.5**|
> |Fully Random|43.3|14.1|15.0|12.4|
> |AR (Left-to-right)|78.0|27.9|34.9|34.5|
>
> **Why random AO fails.** In a dLLM, each token's prediction depends on its surrounding context. A fully random order forces the model to resolve tokens with almost no local context (e.g., predicting token 30 when only token 1 has been decoded). This causes **structural incoherence** rather than merely incorrect reasoning. Below is a representative response from LLaDA under random order:
>
> ```
> 3. **Third Month:**
>    - The number of downloads reduced by 30% compared to the second month.
>      \[
>      180 - (   \times 180 - (   - 180) = 180 - 0.30 \times 180 = 54 \text{ downloads}
>      \]
> ```
>
> LaTeX expressions are corrupted with missing operands and garbled structures (e.g., `180 - ( \times 180`), illustrating a fundamental generation failure rather than a simple reasoning error. This incoherence makes random AO hard to get valid positive reward signals and performs significantly worse than AR order in RL training:
>
> |Method|JustGRPO-Random|JustGRPO (AR Order)|
> |---|---|---|
> |GSM8K Acc.|82.2%|**89.1%**|
>
> **Takeaway.** Taken together with the confidence-based results, these findings suggest that AR order may offer a *"sweet spot"* of context for reasoning: unlike random AO, each token is generated with sufficient preceding context to remain coherent; unlike confidence-based sampling, future tokens are not prematurely revealed in ways that could collapse the exploration of alternative reasoning paths.
>
> ---
>
> **[W2] *Sampling methods and learning efficiency comparison.***
>
> Thank you for raising this important concern. We address the concern below.
>
> **Sampling methods & KV cache.** All baselines in Table 1 use low-confidence remasking during sampling. LLaDA-1.5, LLaDOU, and ESPO decode 1 token per step; the remaining baselines (including SPG) decode 2 tokens per step. Though JustGRPO and other baselines in Table 1 are all without KV-cache due to bidirectional attention of dLLM, we do acknowledge that JustGRPO require more forward passes per rollout than multi-token approaches and learning efficiency comparison under a controlled setting is necessary.
>
> **Controlled learning efficiency comparison.** Therefore, to directly address the reviewer’s concern, we compare JustGRPO and SPG* (SPG rerun under the same training/evaluation protocol) under the same hardware budget (16×H100, GSM8K, generation length 256):
>
> |Method|Acc. at ~10h|Acc. at ~20h|Acc. at ~30h|
> |---|---|---|---|
> |SPG*|**86.5%**|86.8%|86.9%|
> |JustGRPO|85.0%|**88.1%**|**89.1%**|
>
> JustGRPO trails slightly early on (85.0 vs. 86.5 at ~10h), but surpasses SPG by ~20h (88.1 vs. 86.8) and continues improving, while SPG plateaus. This suggests that the gain is primarily associated with better learning dynamics from AR-order rollouts, rather than the sampling-side compute.
>
> **Further acceleration.** Beyond the controlled comparison above, we also introduce JustGRPO-Fast (Appendix D), which restricts gradient computation to the top-25% high-entropy tokens, motivated by our finding that reasoning is disproportionately steered by these forking tokens (§3.2):
>
> |Method|Acc. at ~10h|Acc. at ~20h|Acc. at ~30h|
> |---|---|---|---|
> |JustGRPO-Fast|87.1%|89.8%|89.8%|
>
> JustGRPO-Fast reaches 87.1% in only ~10h (already surpassing SPG's final accuracy) and 89.8% at ~20h, indicating further room for efficiency improvements in future work.
>
> We will incorporate this efficiency analysis into Section 5 of the main text in the revised manuscript.

---

> > ### Author Rebuttal · Reviewer_eiGf · 2026-04-03
> >
> > Thanks for the rebuttal and new experiments. It adequately addressed my questions and it’d be nice if the new experiments will be added to the final paper. And I think it’s important to clarify the sampling difference in main table, as sampling compute determines the quality of the rollout and RL performance.  I remain my positive score.

---

> > > ### Author Response · Authors · 2026-04-06
> > >
> > > We thank the reviewer for confirming that the concerns have been fully resolved and for the positive assessment. We will incorporate the new experiments (random AO baseline and wall-clock efficiency comparison) into the main text of the revised manuscript, and clarify the sampling differences in the main table as suggested. We appreciate the reviewer's constructive feedback, which has meaningfully strengthened the paper.

---

### Official Review · Reviewer_UjZt · 2026-03-15

**Soundness:** 4
**Presentation:** 3
**Significance:** 4
**Originality:** 3
**Overall Recommendation:** 5
**Confidence:** 3

**Summary:**

This paper considers whether the order flexibility in the generative part of discrete diffusion models for text is actually useful or not.  Expanding the space of possibilities beyond left-to-right would seemingly only be helpful, but due to collapse, the argument here is that that is not the case.  For general reasoning tasks (rather than unusual ones such as Sudoku), the paper finds that arbitrary order generation may
limit reasoning capabilities of LLMs, since they exploit this order flexibility to bypass high-uncertainty tokens that are important for exploration in problem solving.  Instead, as part of the inference-compute scaling, it is better to use standard GRPO-type techniques with left-to-right ordering.  This is demonstrated in several experiments using LLaDa and Dream, and state of the art performance is demonstrated on four standard reasoning benchmark tasks.

**Compliance With Llm Reviewing Policy:**

Affirmed.

**Key Questions For Authors:**

1. The authors consider two extremes, fixed order and arbitrary order.  Are there any intermediate orderings that would be compelling to consider?  If so, how would one optimize that intermediate level of ordering information?

2. As noted above, drawing on classical information-theoretic arguments going back e.g. to Varshney and Goyal, might be helpful in arguing for the entropy mechanism.  What do the authors think?

3. I wonder how this justGPRO approach works for standard LLM benchmarks, rather than reasoning tasks.  Would be good to verify if anything is lost.

4. Can one characterize the computational savings in using justGPRO as compared to the more complicated approaches?  That would be another potential advantage of the proposed approach.

**Limitations:**

The impact statement is fine.

**Strengths And Weaknesses:**

In some sense, the central argument within the paper is that simplicity is better than one might expect, and therefore the technical development also seems to espouse this simplicity.  As such, as far as I can tell, the work is technically sound both in terms of the basic theoretical development and in terms of the important experimental work.  If there is some weakness in the technical development, it is that there isn't so strong a justification provided for looking at entropy rather than some other explanatory performance metric; perhaps this can  be mitigated by drawing on some classical information-theoretic arguments about order / not caring about order that may go back to [Varshney and Goyal, "Toward a source coding theory for sets", IEEE DCC, 2006].

The paper is well-written and well-structured.  This is certainly a strength.  For this reader, having the related work be after the intro (rather than near the end), however, would have helped in understanding.

There is strong interest in the community around diffusion-based large language models, especially due to their efficiency, and their growing efficacy.  There hasn't been too much work yet on reasoning with discrete diffusion models, though certainly some, so pushing in this direction is certainly compelling.

The originality of the finding is, in some sense, that originality is not the best approach: that classical approaches are better than novel ones that expand the space of possibilities.

---

> ### Author Rebuttal · Authors · 2026-03-28
>
> We sincerely thank the reviewer for the valuable comments and thoughtful questions, which have greatly helped us improve our work. We address your questions as follows.
>
> ---
>
> **[W1] *Related work placement***
>
> We agree this is a helpful presentation suggestion. We will move the related work section right after the introduction in the revision.
>
> ---
>
> **[Q1] *Are there intermediate orderings between fixed AR and fully arbitrary order? If so, can we optimize them?***
>
> **Yes, and we have systematically studied them.** In practice, dLLMs do not generate in a fully arbitrary order. The standard decoding protocol is *semi-autoregressive*: the sequence is partitioned into blocks of size $B$, and within each block, the model adaptively selects which tokens to unmask based on confidence. The block size $B$ thus directly controls how much flexibility the model has in decoding order: $B=1$ recovers pure AR order; $B=N$ approaches the fully arbitrary case. In Appendix B.3 (Figure 10), we sweep $B \in \{8, 32, 128\}$ alongside $B=1$ (AR) on the HumanEval dataset:
>
> | Block Size $B$ \ Number of Samples $k$ | 1 | 8 | 32 | 128 |
> | - | - | - | - | - |
> | 1 | 34.5 | **59.5** | **72.9** | **83.0** |
> | 8 | 40.4 | 54.7 | 64.1 | 73.3 |
> | 32 | **40.5** | 52.6 | 61.2 | 67.1 |
> | 128 | 40.2 | 51.0 | 57.7 | 62.9 |
>
> The results reveal a consistent trend: **as block size decreases, Pass@k (k>1) improves**, with AR order ($B=1$) as the optimal endpoint. This connects directly to our entropy degradation mechanism (§3.2): larger blocks allow more future tokens to crystallize before high-entropy logical forks are resolved, making it more likely to collapse the branching point retroactively. We will make this discussion more visible in our revised manuscript.
>
> **Could one optimize the intermediate level?** Our results suggest the optimum consistently lies at or near the AR end for general reasoning. Whether task-specific sweet spots exist (e.g., constraint satisfaction where limited lookahead helps) is an interesting future direction.
>
> ---
>
> **[Q2] *Could classical information-theoretic arguments (e.g., Varshney & Goyal, 2006) strengthen the entropy mechanism?***
>
> We thank the reviewer for this insightful connection. Varshney and Goyal (2006) established that the information in an ordered sequence decomposes into content (values) and ordering, with log n! bits attributable to ordering alone. While their entropy is defined at the sequence level (not the token level), the conceptual parallel is instructive: arbitrary-order generation in dLLMs effectively treats the reasoning chain more like a "set" than a "sequence," discarding ordering information that is critical for exploration. Our token-level entropy degradation can be viewed as a local manifestation of this broader principle — at each forking point, the decision entropy that would distinguish different reasoning branches is suppressed when the model is free to resolve context before confronting the fork.
>
> We are happy to incorporate this connection in our revision and cite the suggested reference.
>
> ---
>
> **[Q3]: *How does JustGRPO perform on standard (non-reasoning) LLM benchmarks?***
>
> **Empirically, JustGRPO preserves general capabilities.** We evaluate our JustGRPO checkpoint on four standard non-reasoning benchmarks:
>
> ||**MMLU**|**MMLU-Pro**|**HellaSwag**|**ARC-C**|
> |-|-|-|-|-|
> |Base Model|65.5|**37.0**|74.6|**88.5**|
> |JustGRPO|**65.8**|36.7|**74.8**|87.5|
>
> Overall general capabilities are largely preserved, with small task-dependent fluctuations.
>
>
> Intuitively, this is expected: **the AR formulation in JustGRPO acts purely as a training-time scaffold for RL exploration, rather than a structural constraint on the model.** The model architecture, attention pattern, and pretrained knowledge remain largely untouched. As a result, JustGRPO improves *how the model explores reasoning trajectories* during RL, without interfering with *how it encodes and expresses general knowledge*. We will include these results and a brief discussion in the final version.
>
> ---
>
> **[Q4]: *Can one characterize the computational savings of JustGRPO compared to more complicated approaches?***
>
> Thank you for raising this point. Appendix D (Fig. 11) provides a direct wall-clock efficiency comparison. JustGRPO surpasses the peak accuracy of ESPO (a representative baseline with ELBO approximation) in **less wall-clock time and continues to improve**. Furthermore, our JustGRPO-Fast variant, which restricts gradient updates to the top-25% high-entropy tokens, **achieves even faster convergence**. We believe the efficiency advantage comes from more effective RL exploration and more accurate credit assignment, which we will elaborate on in our revision.

---

> > ### Author Rebuttal · Reviewer_UjZt · 2026-04-03
> >
> > I thank the authors for their further clarifications.  It would be great if these points can indeed be incorporated into the next version of the paper, hopefully in prominent/obvious places to help the reader.

---

> > > ### Author Response · Authors · 2026-04-06
> > >
> > > We sincerely thank Reviewer UjZt for the positive and constructive feedback throughout this process. We are glad that our responses have fully addressed the concerns raised. As promised, we will incorporate the suggested changes in the revised manuscript, including (1) moving the related work section after the introduction, (2) adding the information theoretic connection to Varshney & Goyal (2006), (3) including the general capability benchmarks, and (4) making the block size analysis more prominent. We appreciate the reviewer's time and valuable suggestions, which have meaningfully improved the paper.

---

### Decision · Program_Chairs · 2026-04-30

**Decision:**

Accept (spotlight)

**Comment:**

The promise of diffusion models has been in the flexibility they offer in terms of the order in which the  tokens are produced. This paper challenges this intuition: it argues that this very flexibility can mean the model skips "high uncertainty" tokens that are necessary for exploration. With this, the paper questions RL approaches for diffusion models, instead proposing a simple left-to-right GRPO-based method.

This is a very interesting and valuable hypothesis, and the reviews are unanimously supportive of it, and how the paper has tested it. All reviewers praise its impact: The work can provide an important push for reasoning in dLLMs (`UjZt`), is "a fresh angle on how to do sampling" and "a strong message to the dLLM + RL community" (`eiGf`) and could be valuable for designing more efficient and principled dLLM + RL methods (`BMXv`). The reviewers believe the work is sound both theoretically and experimentally (`UjZt`), the theoretical analysis is thoughtful and experimentally well-supported (`BMXv`).


Initially, concerns were raised by `eiGf` and `BMXv` about unfair/incomplete baselines, the fact that the proposed method is advantaged in terms of compute and missing experimental details (e.g., number of decoding steps).  The authors response appears to have resolved these criticisms.

Overall, we think the ideas in this paper provide a valuable direction for the diffusion community to dwell on, and will help ground the discussion surrounding the pros and cons of diffusion and autoregression.  Therefore, we are happy to recommend acceptance of the paper.